# The NGF R100W Mutation, Associated with Hereditary Sensory Autonomic Neuropathy Type V, Specifically Affects the Binding Energetic Landscapes of NGF and of Its Precursor proNGF and p75NTR

**DOI:** 10.3390/biology12030364

**Published:** 2023-02-25

**Authors:** Sonia Covaceuszach, Doriano Lamba

**Affiliations:** 1Istituto di Cristallografia, Consiglio Nazionale delle Ricerche, 34149 Trieste, Italy; 2Consorzio Interuniversitario “Istituto Nazionale Biostrutture e Biosistemi”, 00136 Roma, Italy

**Keywords:** NGF proNGF, p75NTR, pain, HSAN V, R100W mutation, molecular dynamics simulation

## Abstract

**Simple Summary:**

A point mutation in the Nerve Growth factor gene (leading to the amino acid substitution R100W), causing Hereditary Sensory and Autonomic Neuropathy type V, a condition that primarily affects the sensory nerve cells, whose principal function is to transmit information about sensations, such as pain. Indeed, NGF not only mediates the development and survival of sensory neurons by binding TrkA and p75NTR receptors, but it also plays a role in pain sensation. It is worth noting that the NGF precursor is known to be a biologically active ligand with opposite physiological functions to those of its mature counterpart, and that NGF R100W mutation has been shown to determine an unbalance in the proNGF/NGF levels. Thus, the aim of this work is to elucidate the impact of the R100W mutation on the interactions of p75NTR with the precursor and the mature NGF to unveil the molecular determinants that trigger their different physiological and pathological outcomes. Computer simulations of these complexes allowed us to portray the energetic landscapes and the conformational plasticity, gaining insights into the structural basis of the molecular mechanisms beyond the clinical manifestations of HSAN V patients.

**Abstract:**

Nerve Growth Factor (NGF), the prototype of the neurotrophin family, stimulates morphological differentiation and regulates neuronal gene expression by binding to TrkA and p75NTR receptors. It plays a critical role in maintaining the function and phenotype of peripheral sensory and sympathetic neurons and in mediating pain transmission and perception during adulthood. A point mutation in the NGFB gene (leading to the amino acid substitution R100W) is responsible for Hereditary Sensory and Autonomic Neuropathy type V (HSAN V), leading to a congenital pain insensitivity with no clear cognitive impairments, but with alterations in the NGF/proNGF balance. The available crystal structures of the p75NTR/NGF and 2p75NTR/proNGF complexes offer a starting point for Molecular Dynamics (MD) simulations in order to capture the impact of the R100W mutation on their binding energetic landscapes and to unveil the molecular determinants that trigger their different physiological and pathological outcomes. The present in silico studies highlight that the stability and the binding energetic fingerprints in the 2p75NTR/proNGF complex is not affected by R100W mutation, which on the contrary, deeply affects the energetic landscape, and thus the stability in the p75NTR/NGF complex. Overall, these findings present insights into the structural basis of the molecular mechanisms beyond the clinical manifestations of HSAN V patients.

## 1. Introduction

Nerve Growth Factor (NGF), the prototype of the neurotrophin family, was originally identified as a key survival factor necessary for the development and differentiation of sympathetic and sensory neurons during embryogenesis. It was eventually shown to play pleiotropic functions in several neural and non-neural populations during adulthood, including differentiation, neuronal survival, synaptogenesis, and modulating synaptic plasticity [1]. NGF exerts these activities by binding to two different classes of receptors, the specific Tropomyosin Receptor Kinase A (TrkA) and the low-affinity p75 NeuroTrophin Receptor (p75NTR) [2,3].

NGF is initially synthesised as a longer precursor, known as proNGF [4], with the pro-domain acting as an intramolecular chaperone to guarantee the proper folding of the mature part, and is then further cleaved by specific intracellular proteases [5] giving rise to the C-terminal mature part [6].

Although NGF is essential for the development of nociceptive primary neurons, it also plays a pivotal role in inflammatory hyperalgesia, modulating nociception in adulthood even in conditions with no apparent inflammatory signs. The intracellular mechanisms so far proposed for heat sensitization are direct phosphorylation and membrane trafficking of TRPV1 by TrkA. Long-lasting sensitizing effects are mediated both by changed expression of neuropeptides and ion channels (Na channels, ASIC, and TRPV1) in primary afferents and by spinal NMDA receptors [7]. Moreover, it has been shown that mechano-hypersensitivity from peripheral NGF involves the sphingomyelin signalling cascade activated via p75NTR, and that a peripheral aPKC is essential for this sensitization [8]. p75NTR contributes to cancer-induced bone pain by upregulating mTOR signalling [9], modulates small fiber degeneration in diabetic neuropathic pain [10], and promotes rheumatoid arthritis and inflammatory response by activating the proinflammatory cytokines [11].

The physiological relevance of the NGF system for pain mediation and modulation [12] is supported by significant genetic evidence in humans. To this respect, Hereditary Sensory and Autonomic Neuropathy type V (HSAN V) is an autosomal recessive genetic disorder associated with a point mutation in exon 3 of the *NGFB* gene, leading to the substitution C661T, i.e., a basic arginine (CGG), into an aromatic tryptophan (TGG) at the position corresponding to residue 100 in mature NGF (R100W) [13]. This mutation leads to a severe reduction of pain perception and is associated with a consequent propensity to multiple body lesions. It is worth noting that this genetic disorder is not associated with mental retardation or cognitive impairment [14], contrary to HSAN IV disorder, which is characterized by mutations in the gene coding for *TRKA* [15,16] that leads to congenital insensitivity to pain accompanied by anhidrosis and intellectual disability. The partly overlapping clinical features of HSAN IV and V are in agreement with the general finding that the NGF–TrkA signalling pathways play a crucial role not only in the development, but also in the adult function, of the nociceptive system [7].

These clinical features suggest that the dysfunction of NGF [13] results in a less severe phenotype than the dysfunction of its receptor, TrkA, does [15]. This is most likely the result of a mutated NGF that is still able to bind and activate TrkA, albeit in a defective manner [17,18], whereas HSAN IV-related mutations in TrkA lead to a more dramatic loss of function [16].

In this contest, it is worth noting that also the NGF precursor, proNGF, was shown to be secreted as a biologically active form, exerting opposite functions in nervous system physiology (such as neuronal survival and synaptic plasticity) with respect to those of the mature counterpart [19]. proNGF binds simultaneously to p75NTR and sortilin receptors; the latter one a member of Vps10p-domain receptors [20]. Interestingly, proNGF, unlike NGF, is unable to support physiological pain sensitization [17]. The NGF R100W mutation has been shown to determine an unbalance in the proNGF/NGF levels [21]. Therefore, it is likely that the increased proNGF levels in humans carrying the HSAN V mutation might trigger an endogenous analgesic, suppressing pain perception. In addition, in vitro binding studies of a panel of R100 mutants highlighted that R100 mutations preserve the interaction of NGF with TrkA, but disrupt the interaction with p75NTR [18]. Intriguingly, this effect takes place only in the context of the mature NGF, since the affinity of proNGF R100 mutants to p75NTR does not differ to that of proNGF WT.

From a structural point of view, the X-ray crystal structures of p75NTR complexes with NGF and proNGF, respectively, show a similar 3D architecture, but they surprisingly differ in their stoichiometries [22,23]. The p75NTR/NGF complex (1:2) is asymmetric with two different binding sites. The observed structural asymmetry of the complex [22] has been ascribed to the flexibility of the NGF loop II (highlighted in pink in Figure 1) [24] that disables NGF’s symmetry-related second p75NTR binding site through an allosteric conformational change.

The symmetric 2p75NTR/proNGF [22] is characterized by the presence of an extra binding site located on the mature part, while the electron density corresponding to the entire pro-domain was poorly traceable. The pro regions of proNGF are mostly disordered, and two hairpin loops II at the top of NGF dimer have undergone more conformational changes in comparison to those of the mature NGF structures, suggesting possible interactions with the pro-peptide.

It is worth nothing that this intrinsically disordered pro-peptide does not show any structural effect on p75NTR, and therefore the formation of the symmetrical complexes is mainly due to the pro-domain effect on the mature part, with the conformation of the loop II being the main difference between these complexes [23]. In this respect, Molecular Dynamics (MD) simulations allowed to examine the energetics and the structural characteristics of these two complexes, as well as of the two forms of NGF upon the removal of p75NTR [25].

We now report explicit water MD simulations of the native, as well as of the R100W mutated 2p75NTR/proNGF and p75NTR/NGF complexes, respectively, aiming to unveil the likely structural rearrangements ascribable to the R100W mutation. The results highlight the structural determinants underlying the discriminating functional impact of the HSAN V mutation on the affinity and on the stability of NGF R100W and proNGF R100W complexes with p75NTR.

## 2. Materials and Methods

The PDBePISA server (Proteins, Interfaces, Structures and Assemblies) [26] allowed us to explore the macromolecular surfaces and interfaces of the crystallographic complexes p75NTR/NGF (PDB_ID 1SG1) and 2p75NTR/proNGF (PDB_ID 3IJ2). The structural and physicochemical properties, i.e., hydrogen bonds and salt bridges, Accessible Surface Area (ASA), Buried Surface Area (BSA), and Solvation energy effect (Δ^i^G) were thoroughly analysed.

*Coot* [27] model building tools were used for processing the crystal structures of the 2p75NTR/proNGF and p75NTR/NGF complexes and to mutate the R residue to W at position 100 in both protomers of hNGF 1SG1 (PDB ID 1SG1 chain A: aa 11–115; chain B: aa 9–116) and mouse proNGF (PDB ID 3IJ2 chain A and B: aa 8–117). The missing residues of hNGF, i.e., loop III (aa 61–66), in the crystallographic structure PDB ID 1SG1 were modelled as previously reported [28].

In either of the crystal structures of the 2p75NTR/proNGF and p75NTR/NGF complexes, the residues R100 are localized at the surface in both the NGF or proNGF protomers, and they are exposed. In the p75NTR/NGF structure (PDB ID 1SG1), the ASA values for the R100 residues in the NGF A and B protomers are of 65.4 Å^2^ and 35.8 Å^2^, respectively.

In the 2p75NTR/proNGF structure (PDB ID 3IJ2), the ASA values for the R100 residues in the proNGF A and B protomers are 73.13Å^2^ and 73.16 Å^2^, respectively. Therefore, the R100W mutation did not cause any steric hindrance.

Prior to molecular dynamics simulations, the mutated structures of the complexes were energy minimized.

We further evaluate and discuss the effects of this mutation on the unbound proteins (see below).

MD simulations were performed on the four complexes each solvated in a water box using the GROMACS 4.5 (Groningen Machine for Chemical Simulation) package [29] in conjunction with the Amber99SB force field. The simple point charge model was used to represent the water molecules. The protonation state of the ionizable groups of both the p75NTR and proNGF/NGF proteins was chosen according to a pH of 7.0, and an appropriate number of counter ions were added to achieve charge neutrality in the system. After energy minimization by using a steepest descent algorithm, during the equilibration dynamic period, the system was thermostated to a temperature of 300 K and maintained at a pressure of 1 bar. Starting with these equilibrated structures, MD production runs of 100 ns in duration were performed. All the simulations were performed at 1 atm and 300 K by coupling them to an external heat and an isotropic pressure bath. The impact on the main structural features of the complexes after the emergence of the R100W mutation were analysed by calculating: (1) the root-mean-square deviations (RMSD), (2) the radii of gyration (R_g_), and (3) the distance between the p75NTR—proNGF/NGF centres of mass (COMs) by using GROMACS “gmx rmsd”, “gmx gyrate”, and “gmx distance” tools, respectively.

The GROMACS “gmx rmsf” command line tool allowed us to estimate the individual backbone root-mean-square fluctuation (RMSF) of each protein. The Solvent Accessible Surface Area (SASA) using the GROMACS “gmx sasa” script allowed us to estimate the global stability of the protein–protein complexes. The number of hydrogen bonds and pairs within 0.35 nm were analysed by the GROMACS “gmx hbond” tool.

A descriptive statistical analysis, as implemented in Microsoft Office Excel 10 (Microsoft Corporation, Redmond, WA, USA), was employed to calculate the mean of the obtained values of the analysed parameters as a measure of central tendency, and the standard deviations of the former values were computed as a measure of dispersion. The same software was used to produce all the graphs.

## 3. Results

### 3.1. Impact on the Main Structural Features of the p75NTR/NGF and 2p75NTR/proNGF Complexes after the Emergence of the R100W Mutation

In order to disclose the impact of the R100W pathological mutation on the stability and dynamical behaviour of the complexes of NGF and proNGF with p75NTR, all the p75NTR complexes with WT and mutant NGF or proNGF were used in the 100 ns all-atom MD simulation runs.

At first, the effects of the HSAN V mutation were evaluated by a series of 100 ns all-atom MD simulation runs of the unbound states of NGF and proNGF, both WT and R100W. As shown in Appendix A, by comparing the root-mean-square fluctuations of the two protomers of each molecule, the R100W mutation does not introduce steric hindrance and does not affect the structural features of the neighbouring residues.

The stability of the complexes was explored in terms of the RMSD over the simulation frame. Typically, RMSD is used as a standard measurement for the structural distances between coordinates in order to infer the extent of deviation for a group of atoms relative to their reference structures [30]. The RMSD values display how much the conformations of these groups of atoms change during the simulation, providing a good indication of complex stability [31]. Thus, we calculated the RMSD plots both for the two complexes and the individual subunits.

By comparing the RMSD profiles of the p75NTR/NGF complex (Figure 2a), the R100W mutation induces huge variations in the RMSD values during the 100 ns MD simulation that resulted in an increased mean value (RMSD = 3.69 ± 0.63 Å) compared to that of the complex with the WT protein (RMSD = 2.81 ± 0.47 Å), as reported in Table 1. The same behaviour is shown by both the NGF (Appendix A) and p75NTR (Appendix A). In details, the mean value for the NGF increases from 1.77 ± 0.19 Å for the WT protein to 2.52 ± 0.19 Å for the R100W mutant and, in accordance, the mean value of p75NTR in the complex with NGF WT is 3.77 ± 0.46 Å, while the corresponding value for p75NTR in the complex with NGF R100W is 4.68 ± 0.64 Å (Table 1).

These findings point to a decreased stability of the asymmetric p75NTR/NGF WT complex as a consequence of the R100W mutation. On the contrary, the RMSD profiles of the symmetric 2p75NTR/proNGF WT complex (Figure 2b) do not show significant differences, with RMSD variations that are more similar to the ones observed for the asymmetric p75NTR/NGF WT complex, with them being 3.34 ± 0.42 Å and 3.43 ± 0.40 Å for the p75NTR complexes with proNGF WT and proNGF R100W, respectively (see Table 2). In accordance, the subunits do not show variations also, with the mean values for proNGF WT and R100W being 1.79 ± 0.24 Å and 1.81 ± 0.18 Å, respectively (Appendix A). Regarding the two subunits of p75NTR, while the mean values of the chains X of the complex with proNGF WT and R100W are very similar, i.e., 3.77 ± 0.46 Å and 3.76 ± 0.42 Å, respectively (Appendix A), the value of chain Y of the complex with the mutants slightly decreases respect to that of the complex with the WT, with them being 3.53 ± 0.36 Å and 3.78 ± 0.37 Å for p75NTR_Y in the complexes with proNGF R100W and WT, respectively (Appendix A).

Therefore, the R100W mutation deeply affects the p75NTR/NGF complex, whereas it does not have an impact on the stability of the 2p75NTR/proNGF complex.

These findings are further confirmed by the resulting R_g_ values and the COM-COM distances of the complexes of NGF and proNGF with p75NTR throughout the duration of the simulation.

Generally, low R_g_ values imply sustained stability and compactness of the investigated complexes during the MD simulation. This parameter is defined as the mass-weighted RMSD for a group of atoms relative to their common mass centre. Thus, the structure stability within a valid MD simulation is correlated to R_g_ tones, reaching a plateau around the average values [32].

In the presented study, the R_g_ analysis appears to ensure the preferential stability of the p75NTR/NGF WT complex that was previously investigated by the RMSD trajectory analysis.

In detail, the profile of the R_g_ of the p75NTR/NGFR100W complex is characterized by an increasing trend with consistent variations and a mean value of 26.63 ± 0.22 Å. The mean R_g_ value of the p75NTR/NGF WT complex is significantly lower (i.e., 26.16 ± 0.13 Å) and slightly differs from the value at the start of the simulation (Figure 3a). Conversely, the 2p75NTR/proNGF WT and 2p75NTR/proNGF R100W complexes (Figure 3b) show a similar trend in their R_g_ profiles that oscillate around very close mean values of 30.13 ± 0.21 Å and 30.00 ± 0.18 Å, respectively.

A similar opposite behaviour can be observed comparing the COM-COM distances of the WT and R100W mutants in the p75NTR/NGF and 2p75NTR/proNGF complexes (Figure 4).

Indeed, the COM-COM distances of the p75NTR/NGF WT complex became smaller during the simulation, with small variations and a mean value of 18.38 ± 0.75 Å, which is significantly lower than the mean COM-COM distances of the complex between p75NTR and NGF R100W (i.e., 20.16 ± 0.58 Å), whose profile shows remarkable oscillations (Figure 4a). Instead, the COM-COM distances in the p75NTR/proNGF complexes show similar behaviours for the WT and the R100W mutant, with there being mean values of 18.78 ± 0.83 Å and 18.55 ± 0.56 Å between the chain X of p75NTR and proNGF WT and R100W, respectively, and 18.59 ± 0.52 Å and 19.16 ± 0.50 Å between the chain Y of p75NTR and proNGF WT and R100W, respectively. It is also interesting to note that the fluctuations shown in the COM-COM distances between proNGF WT and both chains of p75NTR (Figure 4b,c) seem to be slightly more pronounced compared to those of the proNGF R100W complexes.

All of these analyses support the hypothesis of a differential impact of the R100W mutation on the stability of p75NTR/NGF and p75NTR/proNGF complexes, resulting in the destabilization of the p75NTR/NGF interaction, while proNGF binding to p75NTR seems to be affected to only a small extent.

### 3.2. Variations in Conformational Flexibility and Their Effect on the Interface Surfaces of the p75NTR/NGF and 2p75NTR/proNGF Complexes after the Emergence of the R100W Mutation

In order to gain a deeper insight into the stability of the complexes after the emergence of the R100W mutation, the RMSF was estimated for each complex throughout the whole MD simulation. RMSF estimates the time evolution of the average deviation for each residue from its reference position within the minimized starting structures [33]. Adopting an ∆RMSF cut-off value of 0.1 Å was relevant in order to estimate the significant change within the structural movements, where residues with >0.1 ∆RMSF values are considered to show significant variations in their mobility [34,35].

When we were analysing the RMSF of each chain of the two complexes (Figure 5), it is worth noting that the flexible regions of the two protomers composing NGF were quite different (Figure 5a,c), while in proNGF (Figure 5b,d) and p75NTR (Figure 5f,g) the RMSF profiles were quite similar.

The highly flexible regions in both the proNGF WT and NGF WT complexes with p75NTR are mainly mapped at the level of the four loops and at the amino and carboxy termini. Loop V of the chain B of NGF in the p75NTR/NGF WT complex shows an increased RMSF with respect to that of chain A, as well as with respect to those of both the proNGF chains in the 2p75NTR/proNGF WT complex. Loop II is less flexible in the chain A of the p75NTR/NGF WT complex with respect to that of chain B, characterized by a higher RMSF, and those of both the proNGF chains in the 2p75NTR/proNGF WT complex, confirming its role in the different binding modes of NGF and proNGF to p75NTR.

The R100W mutation induces an overall increased RMSF along both chains of NGF in the p75NTR/NGF R100W complex, with the only exception being loop II. A similar behaviour characterized the two protomers of proNGF in the 2p75NTR-proNGF R100W complex. A stabilization effect by the R100W mutation can be highlighted in some portions of loop I and loop II in the 2p75NTR-proNGF R100W complex.

A related effect can also be seen in the RMSF profile of p75NTR in the complex with NGF, with only two small patches that seem to be stabilized by the R100W mutation, i.e., residues 6–12 and 27–33 (Figure 5e). A quite different behaviour is observed for the two chains of p75NTR in the 2p75NTR/proNGF complex (Figure 5f,g). Overall, the RMSF in the complex with proNGF WT and R100W are closer along both p75NTR chains and in several regions the mutation seems to stabilize the complex decreasing the RMSF of p75NTR residues. Regarding the NGF, all the contact residues of the two protomers, except for a couple of amino acids (aa) at the end of loop II of chain B, show a significantly higher RMSF after the emergence of the R100W mutation. On the contrary, the RMSD values of both protomers of proNGF do not dramatically change after the emergence of the mutation, and in addition, a stabilizing effect nearby the mutation can be seen in loop V of both the proNGF protomers.

Moreover, focusing on the residues that are involved in the contact interfaces according to [26], and thus responsible for the stability of the complexes (highlighted by violet dots in Figure 5), it can be noted that the RMSF for all the contact residues of p75NTR with NGF show an increase after the emergence of the R100W mutation. Instead, the RMSFs of both the p75NTR chains are very similar to those of the WT and the R100W in the 2p75NTR/proNGF complexes, with some contact regions that are stabilized as result of the mutation.

To investigate this opposite behaviour, we took into account the variation in the RMSF values of the contact residues after the emergence of the R100W mutation (Appendix A). We focused on the residues involved in hydrogen bonds and salt bridges formation and on those that are the major determinant of the formation of the two complexes on the basis of the related ∆^i^G.

Overall, the R100W mutation caused a significant increase in the RMSF of the majority of NGF and p75NTR contact residues (Appendix A), namely 25 aa, with higher RMSF on a total of 25 aa for both NGF protomers and 37 aa out of 43 aa for p75NTR, respectively. In detail, in the p75NTR/NGF complex R100W mutation caused increased fluctuations in five out of the seven residues involved in hydrogen bonds and salt bridges (Appendix A); only W21 and Y52 show a slight decrease in the RMSF, even if none of them contribute significantly to the ∆^i^G (being 0.89 kcal/mol and 0.54 kcal/mol, respectively). From a thermodynamic point of view, the 14 residues (with only one exception) that mainly contribute to the binding energy with negative values of ∆^i^G are all characterized by a significant increase in their RMSF values after the emergence of the R100W mutation. In accordance, the RMSF values of all 12 contacting residues of p75NTR involved in hydrogen bonds and salt bridges (Appendix A) increase because of the R100W mutation, and all the 16 residues carrying negative ∆^i^G show larger fluctuation after the emergence of the mutation. Regarding the impact on the total binding energy (being −3.77 kcal/mol and −2.45 kcal/mol for the NGF protomers and p75NTR, respectively), residues with higher fluctuations after the emergence of R100W account for −3.71 kcal/mol and −2.37 kcal/mol for the NGF and p75NTR, respectively, making the contribution negligible to the binding energy by the few residues, with a reduced RMSF for both the binding partners (i.e., −0.06 kcal/mol and −0.08 kcal/mol for NGF and p75NTR, respectively). All these considerations are in agreement, pointing to a destabilizing effect of this interaction due to the R100W mutation.

As the global effect of the R100W mutation on the 2p75NTR/proNGF complex (Appendix A) is of concern, a significant decrease was observed in the RMSF values of the key residues involved in stabilizing non-covalent interactions and that are mainly responsible of the strength of the interaction. In particular, lower RMSF values are associated with the R100W mutation in 10 of the 14 residues involved in hydrogen bonds and salt bridges of the two chains of p75NTR, while on the side of the two proNGF protomers, only 3 residues out of a total of 13 show a significant increase in the RMSF values. As a result, 21 residues on 34 p75NTR that contribute to the overall binding energy are characterized by a reduction of their RMSF values, while 12 residues on 25 of the proNGF protomers show higher fluctuations. Regarding the contribution on the total binding energy (being −7.50 kcal/mol and −4.77 kcal/mol for proNGF protomers and the two chain of p75NTR, respectively), those residues with decreased RMSFs after the emergence of the R100W mutation account for −3.96 kcal/mol and −3.11 kcal/mol for the NGF and the two chains of p75NTR, respectively. Therefore, the residues showing increased fluctuations influence only for 47.2% and 34.8% of the total ∆^i^G for proNGF and for the two chains of p75NTR, respectively, with a possible compensatory effect of the contact residues with decreased RMSFs. Indeed, they account for the majority of the binding energy contributors in both the binding partners. These data strongly suggest that HSAN V mutation has an opposite impact on the 2p75NTR/proNGF complex with respect to that on the p75NTR/NGF complex, increasing the strength of the former one, while promoting the destabilization of the latter one.

### 3.3. Global Stability Analysis of the Interacting Surfaces

Solvent Accessible Surface Area (SASA) estimates the molecular surface area accessible to solvent molecules, providing a quantitative measurement of the extent of the protein/solvent interactions [36]. Decreased SASA tones generally imply relative molecular surface shrinkage of the protein–protein complexes by altering the solvent exposed surface charges, yielding, in turn, more compact and stable conformations.

Aiming to investigate the impact of R100W mutation on the p75NTR/NGF complex, the total SASA trajectories of the two complexes were monitored (Figure 6a).

The p75NTR/NGF WT complex showed significantly lower SASA trajectories (i.e., 2280.23 ± 159.15 Å^2^) with respect to those of the p75NTR/NGF R100W complex (i.e., 2657.02 ± 177.08 Å^2^), suggesting a reduction of the intermolecular interaction surface due to the R100W mutation, which is quite a stable effect, taking into account the small fluctuations in the total SASA profile of both complexes (Figure 6a). Analogously, the R100W mutation causes an increase in the hydrophobic SASA from 1086.15 ± 122.35 Å^2^ to 1253.44 ± 106.87 Å^2^ (Figure 7a), as well as of the hydrophilic SASA from 1194.16 ± 109.44 Å^2^ to 1403.65 ± 97.10 Å^2^ (Figure 8a).

Even if the fluctuation in both the hydrophobic and hydrophilic SASA values are larger than those observed for the respective total SASA, this finding confirms the destabilizing effect of the R100W mutation on the p75NTR/NGF complex.

Conversely, the total SASA tones of both the 2p75NTR-proNGF WT and R100W complexes were comparable with a slight decrease in the SASA values for the 2p75NTR-proNGF R100W complex. The total SASA values were 2758.12 ± 283.04 Å^2^ and 2542.33 ± 221.51 Å^2^ for proNGF WT in the complex with the two chains of p75NTR and 2842.15 ± 181.19 and 2558.79 ± 201.21 Å^2^ for proNGF R100W in the complex with the two chains of p75NTR (Table 2), respectively. This finding points to the similar stability of the two complexes, even if there were significant fluctuations in the SASA values during the simulation. The profiles, nevertheless, are shared by all the analysed complexes (Figure 6b). The analysis of the both hydrophobic (Figure 7b) and hydrophilic SASA (Figure 8b) agrees well with the dynamic behaviour of the total SASA (Table 2), even if the decrease in their tones after the mutation emerged is less pronounced.

### 3.4. Analysis of the Evolution of Intermolecular Hydrogen Bonding Network after the Emergnce of the HSAN V Mutation

The stability of the hydrogen bond network interactions between the engaged core protein residues were investigated during the MD simulation. The time-dependent variations in the number of the hydrogen bonds and in the interacting pairs within 3.5 Å were observed, together with their stability during the MD simulation window for calculating the corresponding percentage of time persistence (Figure 9 and Figure 10, respectively).

Regarding the p75NTR/NGF complex, the number of hydrogen bonds (Figure 9a) and of the interacting pairs within 3.5 Ă (Figure 10a) apparently did not seem to be influenced by the R100W mutation (Table 1), with the number of hydrogen bonds being 10.06 ± 3.31 and 11.13 ± 2.28 for the complexes with NGF WT and R100W, respectively. The number of interacting pairs within 3.5 Ă are 5.47 ± 2.66 and 5.78 ± 2.31 for the complexes with NGF WT and the mutant, respectively. It is worthy of note that both the trajectories are characterized by large variations, suggesting that beside the mean number of hydrogen bonds or of interacting pairs, a more significant parameter to be compared is their time persistence, expressed by calculating the percentage of time during the entire simulation. In details, the number of hydrogen bonds covers a range between 0% and 20% and shows a large fluctuation mainly in the first 50 ns of the simulation, especially in the case of the p75NTR-NGF R100W complex. This behavior is reflected by comparing the percentage of time persistence of the number of hydrogen bonds in the p75NTR complexes with NGF WT and R100W, respectively. The corresponding plots show Gaussian profiles with a peak maximum height that is shifted to a lower number and a lower percentage of time persistence after the emergence of the R100W mutation (i.e., 11 hydrogen bonds. lasting for a percentage around 18% of time for the p75NTR/NGF WT complex, compared to 9 hydrogen bonds, lasting for a percentage around 15% of time for the p75NTR/NGF R100W complex).

On the other hand, the number of the interacting pairs within 3.5 Ă also spans over a broad range (between 0% and 19%) and shows large fluctuation during the entire simulation window. Both plots are characterized by an almost overlapping Gaussian profile that has lower peaks after the emergence of the R100W mutation, with a maximum of six interacting pairs within 3.5 Ă, lasting for a time persistence around 17% for the p75NTR/NGF WT complex, compared to seven interacting pairs within 3.5 Ă, lasting for a persistence time around 14% for the p75NTR/NGF R100W complex. All of these findings point to a weakened hydrogen bond interactions network induced by the HSAN V mutation in the p75NTR/NGF complex.

Concerning the symmetric 2p75NTR/proNGF complex, as observed in the case of the asymmetric p75NTR/NGF complex, the R100W mutation does not affect the number of hydrogen bonds (Figure 9b,c) and of the interacting pairs within 3.5 Ă (Figure 10b,c). Indeed, the numbers of hydrogen bonds with p75NTR chain X are 10.94 ± 2.55 and 9.05 ± 2.35 and with p75NTR chain Y 7.46 ± 2.24 and 8.23 ± 2.15 for the p75NTR complexes with proNGF WT and proNGF R100W, respectively. The numbers of interacting pairs within 3.5 Ă with p75NTR chain X are 6.68 ± 2.65 and 5.68 ± 2.27 and with p75NTR chain Y 5.18 ± 2.23 and 5.56 ± 2.07 for the complexes with proNGF WT and the proNGF mutant, respectively (Table 2).

By analyzing the corresponding trajectory of the number of hydrogen bonds for the p75NTR_X/proNGF complex, the plot is characterized by slightly reduced fluctuations compared to those of the asymmetric p75NTR/NGF WT complex. The Gaussian profiles of the corresponding plots of the percentage of persistence time have a peak height maximum that is shifted towards a lower number of hydrogen bonds, but show a higher percentage of time persistence as result of the R100W mutation (i.e., 11 hydrogen bonds, lasting for a persistence time around 16% for the p75NTR_X complex/proNGF WT, compared to 8 hydrogen bonds, lasting for a persistence time of around 20% for the p75NTR/proNGF R100W complex). In the case of p75NTR_Y/proNGF WT complex, the fluctuations are even more reduced, while for the complex with the mutant, an increase in the number of hydrogen bonds can be observed in the last 20 ns of the simulation. The corresponding plots of the percentage of time are characterized by a symmetric Gaussian profile for the p75NTR_Y/proNGF WT complex with a maximum of eight hydrogen bonds, lasting for a persistence time of around 17% as in the case of the p75NTR_Y complex/proNGF R100W. The profile of the latter one is broader and asymmetric, spanning to the range of 9–11 hydrogen bonds and lasting for a persistence time between 17 and 12%.

Concerning the trajectories of the interacting pairs within 0.35 nm, the plots of the complexes with p75NTR_X show a similar range of fluctuations to those of the asymmetric p75NTR/NGF complex. Both the plots display a broad and an almost overlapping Gaussian profile, which peak height increases after the emergence of the R100W mutation, with a shared maximum of five pairs within 3.5 Ă, lasting for a higher percentage of persistence time for the p75NTR_X/proNGF R100W complex with respect to that of p75NTR_X/proNGF WT (26 and 15%, respectively). Instead, the trajectories of the complexes with p75NTR_Y exhibit a narrower range of fluctuations compared to those of the p75NTR/NGF complex. The Gaussian profiles of the percentage of persistence time display a peak height maximum that is shifted to a higher number and to a higher percentage of persistence time after the emergence of the R100W mutation (i.e., six pairs within 3.5 nm, lasting for a persistence time around 19% for the p75NTR_Y/proNGF R100W complex, compared to five pairs within 3.5 Ă, lasting for a persistence time around 18% for the p75NTR_Y/proNGF WT complex). Thus, these observations suggest that the HSAN V mutation not only does not negatively affect the hydrogen bond interaction network in the symmetric 2p75NTR/proNGF complex, but it might, on the contrary, strengthen the non-covalent interactions that mediate the formation of this complex.

### 3.5. Dynamics Analysis of the p75NTR/NGF and 2p75NTR/proNGF Complexes after the Emergence of the R100W Mutation

We finally compared the 10 most populated poses of each simulation, obtained by GROMACS “*cluster.com*” command line tool, to gain information on the relevant collective motions characterizing the asymmetric p75NTR/NGF and symmetric 2p75NTR/proNGF complexes after the emergence of the R100W mutation. As shown in Figure 11, the motions of both NGF and p75NTR increase after the emergence of the R100W mutation, not only at the level of the interaction surface comprising position 100, as expected (Figure 11, lower panel), but also at the level of the second main contact surface that involves the domain at the C-terminal of the extracellular domain of p75NTR.

This unexpected dynamic behavior suggests that R100W mutation is also able to destabilize the asymmetric p75NTR/NGF complex by long range cooperative effects that significantly contribute to disrupt this interaction.

On the contrary, the dynamics analysis of the symmetric 2p75NTR/proNGF complex (Figure 12 and Figure 13) show that, even if the overall motions of both proNGF and p75NTR slightly increase after the emergence of the R100W mutation, the interaction surfaces are not affected especially in the regions involving position 100 (Figure 12 and Figure 13, lower panel).

In conclusion, the present in silico study of the NGF and proNGF complexes with p75NTR after an HSAN V mutation emerges proves that the main effect of this mutation is to lower NGF affinity toward the p75NTR receptor, not only by means of a direct destabilization of the interaction surface at the level of position 100, but also by long range cooperative effects. This it takes place only in the context of the asymmetric p75NTR/NGF complex, whereas the R100W mutation does not interfere with the stability of the interaction between p75NTR and proNGF, providing a structural basis of the molecular mechanisms beyond the clinical manifestations of HSAN V patients.

## 4. Discussion

In this paper, we used water MD simulations to gain insights into the structural determinants underlying the discriminating functional impact of the HSAN V mutation on the affinity and on the stability of NGF complex with p75NTR.

HSANs are a heterogeneous group of eight phenotypically diverse forms of inherited peripheral neuropathies, characterized by mainly sensory, but also variable motor and minimal autonomic dysfunctions [37]. Among them, HSAN V is characterized by the loss of pain perception without any mental retardation. This is the result of the substitution of C to T at nucleotide position 661 (661C > T) in exon 3 of the NGF gene located on chromosome 1p11.2-p13.2, resulting in a missense mutation of tryptophan (W) for arginine (R) at position 221 in the proNGF polypeptide corresponding to residue R100W in mature NGF [13]. At first, aiming to elucidate the functional impact of NGF and proNGF binding to TrkA and p75NTR receptors, we focused on the structural analysis of the X-ray crystal structures of the complexes that were available in the Protein Data Bank. We have previously highlighted [18] that R100 does not participate in the interacting symmetric protein–protein molecular surface with the TrkA receptor (2:2) (PDB_ID 2IFG) [38], and above all, that R100 is buried (90%) in the asymmetric complex p75NTR/NGF complex and is engaged in a salt-bridge protein–protein interaction with residue D75 of p75NTR (PDB_ID 1SG1) [22], largely contributing to the binding affinity of the complex. Due to the negligible yields obtained in attempting the production of recombinant human NGF R100W in an *E. coli* heterologous expression system, the characterization of the in vitro receptor binding properties toward NGF receptors by Surface Plasmon Resonance (SPR) (Table 3) was performed on a panel of single mutants in position 100 (R-> K, E, Q, A, and V) [18], proving that TrkA binding affinity is unaffected, while the R100 mutations greatly reduce the interaction between NGF and p75NTR. In this work, we confirmed that this finding also applies to the NGF R100W mutant (Table 3 and Appendix A). Interestingly, the R100 mutations display only a minor impact on p75NTR binding affinity in the context of the unprocessed hproNGF R100 muteins [17].

proNGF, unlike NGF, is unable to support pain sensitization [17], and taking into account that patients suffering with HSAN V show an imbalance between the NGF and proNGF levels [21], it has been postulated that proNGF might suppress pain perception by acting as an endogenous analgesic.

The unaffected in vitro binding affinity of p75NTR to proNGF cannot be easily reconciled only on the basis of the so far available crystallographic structure of the symmetric p75NTR/NGF complex (PDB_ID 3iJ2) [23]. Indeed, the interaction surfaces of this complex almost completely recapitulate the one of the asymmetric p75NTR/NGF complex with the involvement of R100 residues in the interacting protein–protein molecular surfaces.

To clarify this apparent paradox, we pursued an in silico computational approach previously exploited by Pimenta et al., 2014 [25], which employed a series of 100 ns MD simulations to elucidate the structural features and the binding energetics of p75NTR complexes with NGF and proNGF with (1:2) and (2:2) stoichiometries, respectively. Indeed, MD simulations are widely considered to be an efficacious approach to exploit the stability of intermolecular protein–protein complexes, as well as to investigate their relative dynamic nature, providing detailed information on the energetics of binding interactions and the impact of mutations on protein–protein interactions [39]. Therefore, we first reproduced the previous results [25] to validate our system. Then, we produced MD simulations for the same complexes in which the R100W mutation has been included. The comparative analysis of the resulting trajectories highlighted the postulated differential impact of the HSAN V mutation on the asymmetric p75NTR/NGF and symmetric 2p75NTR/proNGF in terms of (i) the behaviour of all the structural parameters (RMSD, R_g_, and COM-COM distance), (ii) the dynamics of the interacting protein–protein surfaces deciphering the interaction hydrophobic and hydrophilic fingerprints, (iii) the time-persistence of the hydrogen bond networks, and (iv) the overall energetic landscapes (∆^i^G). Indeed, as previously hypothesised, the R100W mutation deeply destabilises p75NTR/NGF (1:2) asymmetric complex, whose RMSD, R_g_, and COM-COM distances increase, but most importantly, the hydrogen bond network is disrupted, and as a consequence, the interacting molecular protein–protein surface is significantly shrunken, and the binding energy highly reduced. Surprisingly, instead, proNGF interaction with p75NTR in the (2:2) symmetric complex with the R100W mutation positively improve the binding affinity.

## 5. Conclusions

In conclusion, all of the outcomes of the present MD simulation are in agreement, pointing to a destabilization effect of the HSAN V mutation, which specifically affect the asymmetric p75NTR/NGF (1:2) complex alone, while the symmetric 2p75NTR/proNGF (2:2) complex is strengthened, allowing us to reconcile the apparent discrepancy between the crystallographic information and the in vitro experimental SPR data. Moreover, elucidating the differential dynamics of the two complexes after the emergence of the R100W mutation allows us to gain novel insights into the differential biological outcomes observed between the mature and proforms of neurotrophins in pain transmission and perception, providing the structural basis for clarifying the molecular determinants triggering the clinical manifestations in patients affected by HSAN V.

## Figures and Tables

**Figure 1 biology-12-00364-f001:**
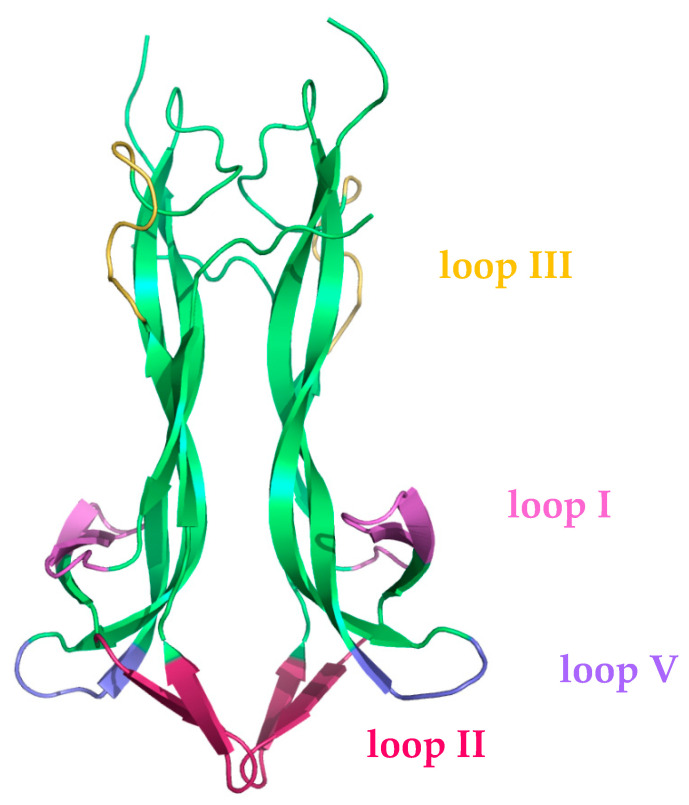
Cartoon representation of hNGF, whose loops are highlighted (i.e., loop I in magenta, loop II in pink, loop III in yellow, and loop V in violet).

**Figure 2 biology-12-00364-f002:**
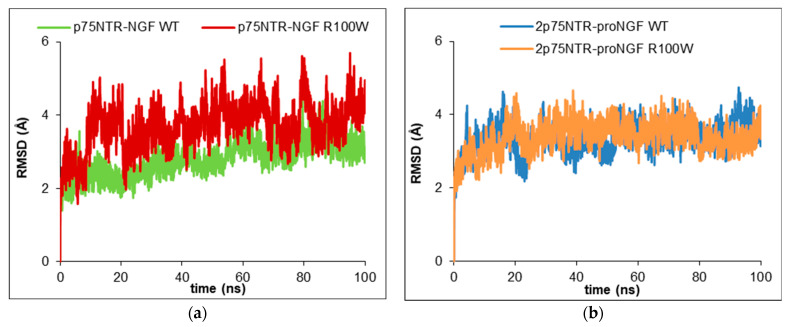
Root-mean-square deviation plots of the complexes: (**a**) p75NTR/NGF and (**b**) 2p75NTR/proNGF.

**Figure 3 biology-12-00364-f003:**
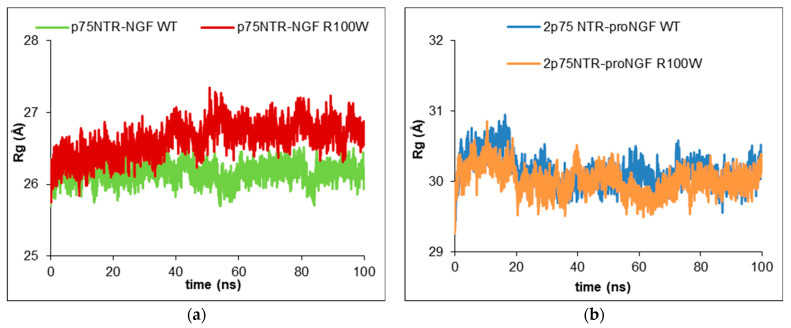
Radius of gyration plots of the complexes: (**a**) p75NTR/NGF and (**b**) 2p75NTR/proNGF.

**Figure 4 biology-12-00364-f004:**
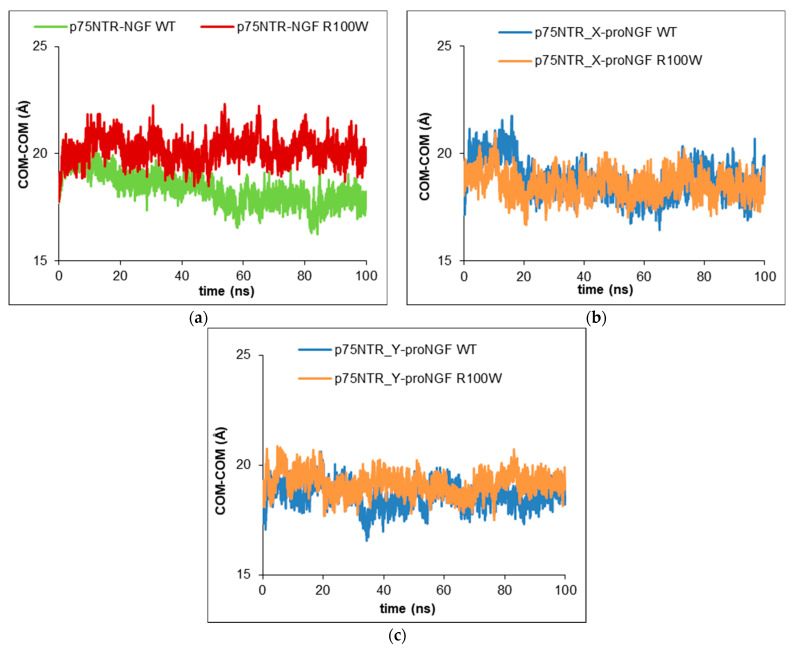
Plots of the centres of mass distance of the complexes: (**a**) p75NTR/NGF, (**b**) p75NTR_X/proNGF, and (**c**) p75NTR_Y/proNGF.

**Figure 5 biology-12-00364-f005:**
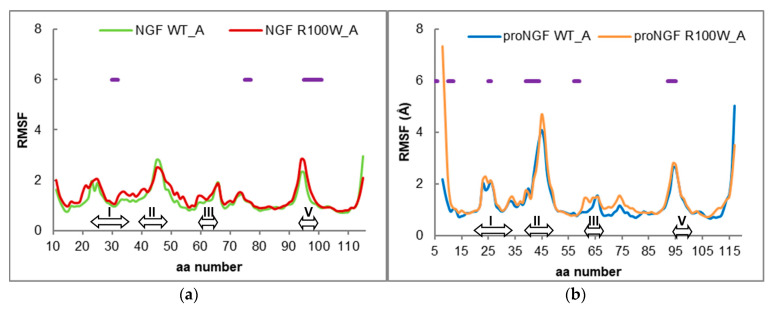
Plots of root-mean-square fluctuation of the backbone of the different chains composing both complexes: (**a**) NGF_A, (**b**) proNGF_A, (**c**) NGF_B, (**d**) proNGF_B, (**e**) p75NTR, (**f**) p75NTR_X, and (**g**) p75NTR_Y (the back arrows near the x axes map the NGF loops, according to [24]; the violet dots in the upper part below the legends highlight the contact interface residues).

**Figure 6 biology-12-00364-f006:**
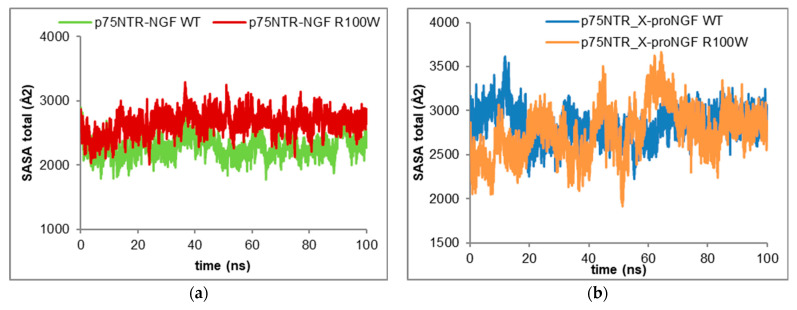
Time-evolution Solvent Accessible Surface Area trajectories of the complexes: (**a**) p75NTR/NGF, (**b**) p75NTR_X/proNGF, and (**c**) p75NTR_Y/proNGF.

**Figure 7 biology-12-00364-f007:**
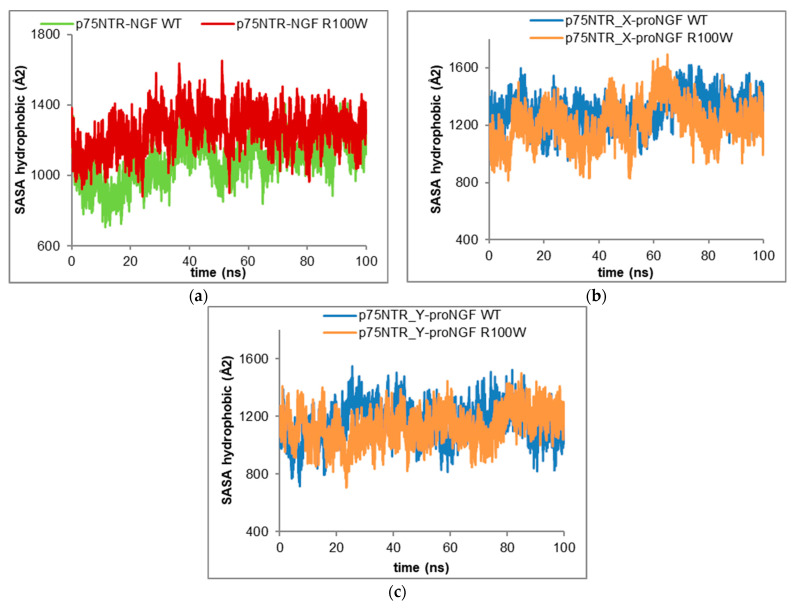
Time-evolution hydrophobic Solvent Accessible Surface Area trajectories of the complexes: (**a**) p75NTR/NGF, (**b**) p75NTR_X/proNGF, and (**c**) p75NTR_Y/proNGF.

**Figure 8 biology-12-00364-f008:**
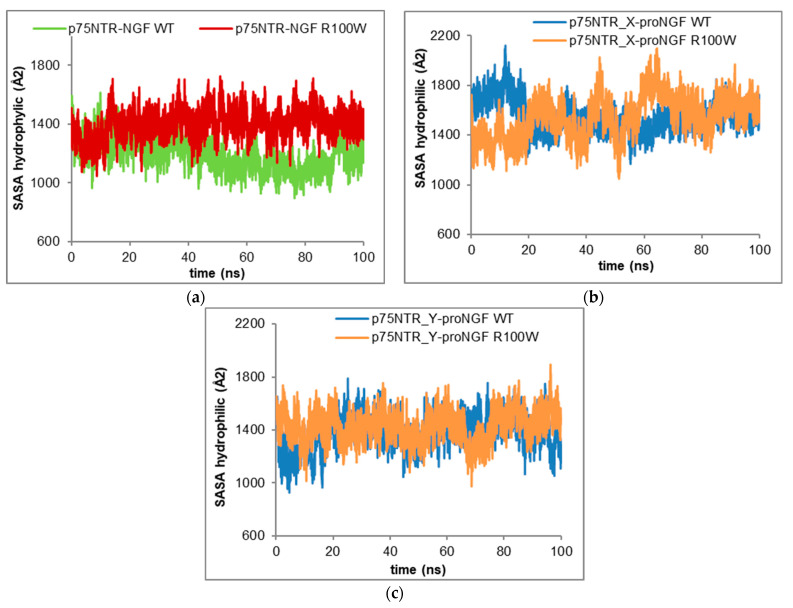
Time-evolution hydrophilic Solvent Accessible Surface Area trajectories of the complexes: (**a**) p75NTR/NGF, (**b**) p75NTR_X/proNGF, and (**c**) p75NTR_Y/proNGF.

**Figure 9 biology-12-00364-f009:**
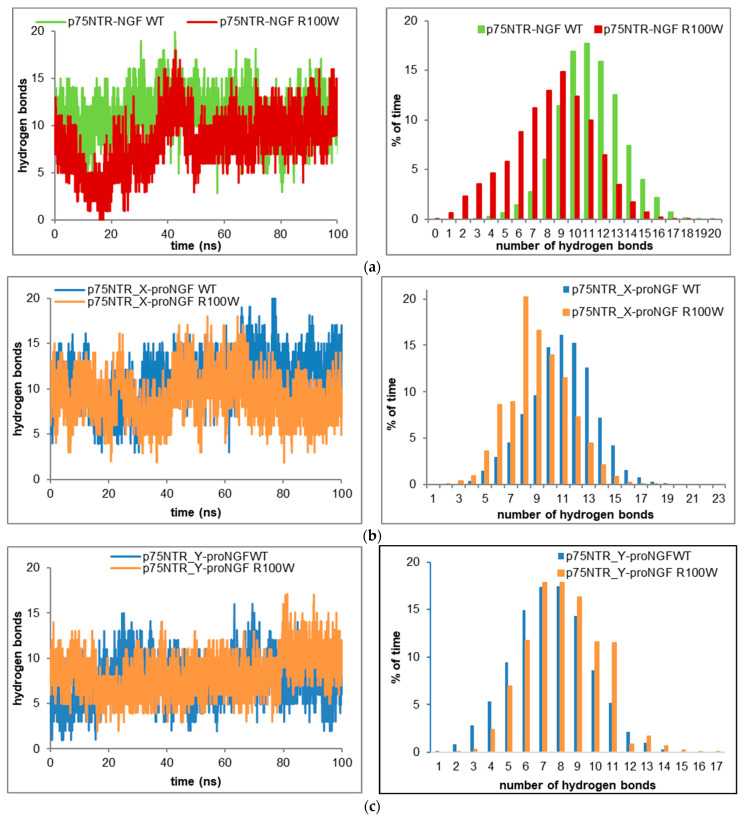
Plots of the number (**left** panel) and percentage of time persistence (**right** panel) of the intermolecular hydrogen bonds in the complexes: (**a**) p75NTR/NGF, (**b**) p75NTR_X/proNGF, and (**c**) p75NTR_Y/proNGF.

**Figure 10 biology-12-00364-f010:**
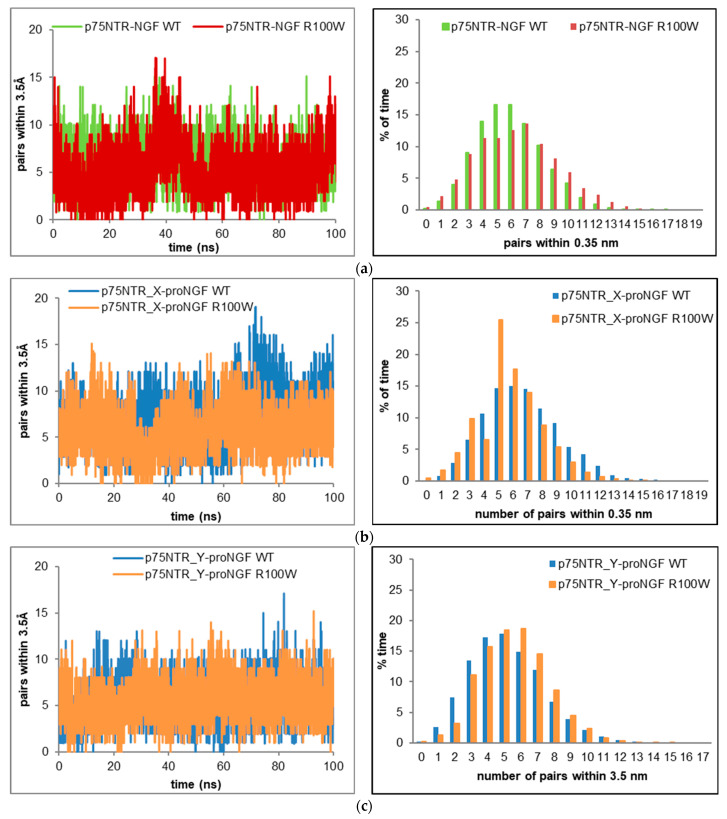
Plots of the number (**left** panel) and percentage of time persistence (**right** panel) of the intermolecular pairs within 3.5 Ă in the complexes: (**a**) p75NTR/NGF, (**b**) p75NTR_X/proNGF, and (**c**) p75NTR_Y/proNGF.

**Figure 11 biology-12-00364-f011:**
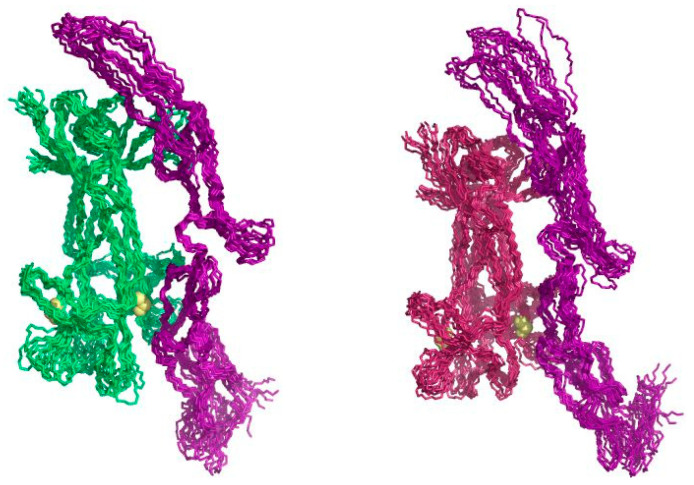
The structural ensembles generated by MD simulations for the complexes of p75NTR (in purple) with (**a**) NGF WT (in green) and (**b**) NGF R100W (in magenta). The bundle of 10 most populated poses along the 100 ns simulation window for the overall complexes (**upper** panel) and the detail of the contact surface involving position 100 (**lower** panel), whose backbone atoms are highlighted by yellow spheres, are shown. Figure produced by Pymol (Schrödinger LLC 2010).

**Figure 12 biology-12-00364-f012:**
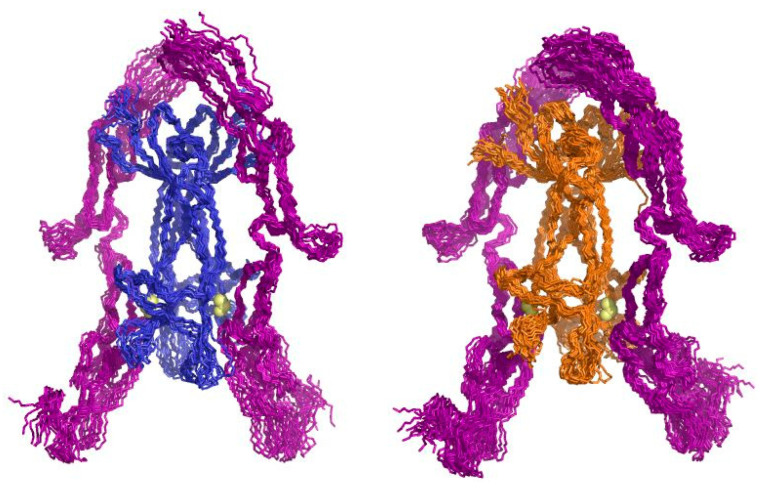
The structural ensembles generated by MD simulations for the complexes of p75NTR (in purple) with (**a**) proNGF WT (in blue) and (**b**) NGF R100W (in orange). The bundle of 10 most populated poses along the 100 ns simulation window for the overall complexes (**upper** panel) and the detail of the contact surface involving position 100 with the chain X of p75NTR (**lower** panel), whose backbone atoms are highlighted by yellow spheres, are shown. Figure produced by Pymol (Schrödinger LLC 2010).

**Figure 13 biology-12-00364-f013:**
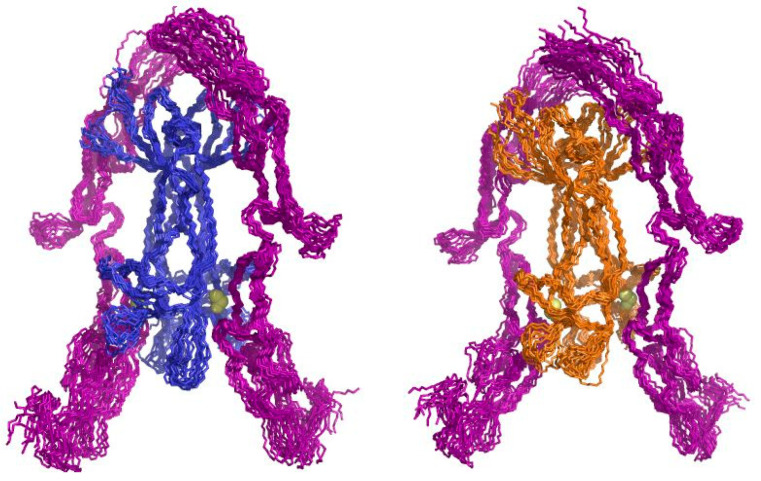
The structural ensembles generated by MD simulations for the complexes of p75NTR (in purple) with (**a**) proNGF WT (in blue) and (**b**) NGF R100W (in orange). The bundle of 10 most populated poses along the 100 ns simulation window for the overall complexes (**upper** panel) and the detail of the contact surface involving position 100 with the chain Y of p75NTR (**lower** panel), whose backbone atoms are highlighted by yellow spheres, are shown. Figure produced by Pymol (Schrödinger LLC 2010).

**Table 1 biology-12-00364-t001:** Summary of the main structural parameters of the p75NTR/NGF complex after the emergence of the R100W mutation over the 100 ns simulation frame.

p75NTR/NGF (PDB_ID 1SG1)	p75NTR/NGF WT	p75NTR/NGFR100W
RMSD (Å)	2.81 ± 0.47	3.69 ± 0.63
RMSD NGF (Å)	1.77 ± 0.19	2.52 ± 0.19
RMSD p75NTR (Å)	3.77 ± 0.46	4.68 ± 0.64
R_g_ (Å)	26.16 ± 0.13	26.63 ± 0.22
COM-COM distance (Å)	18.38 ± 0.75	20.16 ± 0.58
Number of hydrogen bonds between p75NTR-NGF_AB	10.06 ± 3.31	11.13 ± 2.28
Number of pairs within 3.5Å between p75NTR-NGF_AB	5.47 ± 2.66	5.78 ± 2.31
SASA total (Å^2^)	2280.23 ± 159.15	2657.02 ± 177.08
SASA hydrophobic (Å^2^)	1086.15 ± 122.35	1253.44 ± 106.87
SASA hydrophilic (Å^2^)	1194.16 ± 109.44	1403.65 ± 97.10

**Table 2 biology-12-00364-t002:** Summary of the main structural parameters of the 2p75NTR/proNGF complex after the emergence of the R100W mutation over the 100 ns simulation frame.

2p75NTR/proNGF (PDB_ID 3IJ2)	2p75NTR/proNGF WT	2p75NTR/proNGF R100W
RMSD complex (Å)	3.34 ± 0.42	3.43 ± 0.40
RMSD proNGF (Å)	1.79 ± 0.24	1.81 ± 0.18
RMSD p75NTR_X (Å)	3.77 ± 0.46	3.76 ± 0.42
RMSD p75NTR_Y (Å)	3.78 ± 0.37	3.53 ± 0.36
R_g_ (Å)	30.13 ± 0.21	30.00 ± 0.18
COM_COM distance p75NTR_X-proNGF (Å)	18.78 ± 0.83	18.55 ± 0.56
COM_COM distance p75NTR_Y-proNGF (Å)	18.59 ± 0.52	19.16 ± 0.50
Number of hydrogen bonds between p75NTR_X-proNGF_AB	10.94 ± 2.55	9.05 ± 2.35
Number of pairs within 3.5Å between p75NTR_X-proNGF_AB	6.68 ± 2.65	5.68 ± 2.27
Number of hydrogen bonds between p75NTR_Y-proNGF_AB	7.46 ± 2.24	8.23 ± 2.15
Number of pairs within 3.5Å between p75NTR_Y-proNGF_AB	5.18 ± 2.23	5.56 ± 2.07
SASA total p75NTR_X-proNGF (Å^2^)	2842.15 ± 181.19	2758.12 ± 283.04
SASA hydrophobic p75NTR_X-proNGF (Å^2^)	1297.18 ± 102.28	1215.25 ± 139.63
SASA hydrophilic p75NTR_X-proNGF (Å^2^)	1545.06 ± 124.78	1542.36 ± 167.77
SASA total p75NTR_Y-proNGF (Å^2^)	2542.33 ± 221.51	2558.79 ± 201.21
SASA hydrophobic p75NTR_Y-proNGF (Å^2^)	1157.06 ± 121.13	1129.87 ± 112.64
SASA hydrophilic p75NTR_Y-proNGF (Å^2^)	1385.36 ± 126.41	1429.03 ± 114.55

**Table 3 biology-12-00364-t003:** Summary of the SPR derived equilibrium-binding constants **K_D_** (nM) of hNGF and hproNGF and of their mutants in position 100 towards TrkA and p75NTR receptors [17,18].

hNGF—Mutants	TrkA K_D_ (nM)	p75NTR K_D_ (nM)
hNGF	0.94	1.53
hNGF R100W	0.95 ± 0.15	100.00 ± 7.81
hNGF R100E	1.44	125.00
hNGF R100Q	1.63	25.9
hNGF R100K	1.75	2.65
hNGF R100A	1.29	6.44
hNGF R100V	0.65	9.58
**hproNGF—Mutant**	**TrkA K_D_ (nM)**	**p75NTR K_D_ (nM)**
hproNGF	17.80	19.70
hproNGF R100E	20.60	52.50

## Data Availability

Not applicable.

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
