# Peer review of "The NGF R100W Mutation, Associated with Hereditary Sensory Autonomic Neuropathy Type V, Specifically Affects the Binding Energetic Landscapes of NGF and of Its Precursor proNGF and p75NTR"

_biology, 2023, doi:10.3390/biology12030364_

Round 1

Reviewer 1 Report

The manuscript by Covaceuszach and Lamba reports a molecular dynamic study of the interaction of wild-type and of a disease-related mutant (R100W) of NGF/ProNGF with the receptor p75NTR. The topic of the manuscript is of interest as this naturally occurring mutant is responsible of the Hereditary Sensory and Autonomic Neuropathy type V (HSAN V), a pathological state related to a congenital pain insensitivity. The manuscript is clearly written and technically sound, although the timescale of the simulations (100 ns) is somewhat limited considering the nowadays currently available computational resources. Therefore, I think that the manuscript can be considered for publication. There are, however, some important aspects that should addressed upon revision. In particular, I suggest the following analyses and integrations:

1) The authors should clearly report the residues range of the simulated models for both NGF and proNGF. How different are they in terms of sequence? The authors state (line 142) that missing residues in the starting crystallographic structures were modelled. Has this be done for the flexible N-terminal region of proNGF? The authors should provide in the Methods section further details on the R to W replacement. Did it introduce steric strain?

2) The authors frequently relate RMSD and RMSF values with the stability of the systems. RMSD values simply measure the differences with the reference state whereas RMSF values are related to the flexibility that is not necessarily an indicator of instability. For a better understanding of the conformational variations occurring during the simulation, RMSD values could also be reported for the different subunits of the complexes and compared to the global ones. May the large RMSD value observed for the p75NTR/NGF R100W mutant associated in some strain generated by the modeling of mutation in the initial model?

3) The authors use the SASA as one measure of the strength of the complexes. Although somehow related to SASA, the evaluation of the buried surface is probably a better indicator.

4) In the analysis of the hydrogen bonds that stabilize the complex between NGF/proNGF, the evolution of the distance of the partners (H donor and acceptor) as function of time would be informative about the stability of the interaction. This analysis could highlight differences in the stability of the complex of wild-type and mutant proteins.

5) The complexes formed by NGF and proNGF with p75NTR have a different stoichiometry. Is there an interplay between the effect of the mutation and the stoichiometry? Is the effect of the mutation directly caused by the presence/absence of the NGF pro-region or this effect is mediated by the different stoichiometry of the complexes (induced by the pro-region)?

6) In principle, the effect of a mutation on binding also depends on the impact of the mutation on the unbound state(s).  Although minor effects are expected in these systems as the mutation site is surface exposed, this aspect should be commented.

7) No experimental data is available on the affinity of the R100W mutants for the receptor.  Therefore, the authors compare their results with the data available for other mutants. Although not mandatory, they could consider the possibility of performing simulations on the R100E mutant to further validate their approach?

Reviewer 2 Report

1. Statistical tests used to detect significance should be mentioned and the program used to construct graphs and its version should be mentioned in material section'

2- all tables and graphs miss signs of significance

3- all abbreviations should be written  full in figure or table legend
